# Toll-Like Receptor Signaling in the Establishment and Function of the Immune System

**DOI:** 10.3390/cells10061374

**Published:** 2021-06-02

**Authors:** Jahnavi Aluri, Megan A. Cooper, Laura G. Schuettpelz

**Affiliations:** Department of Pediatrics, Washington University School of Medicine, St. Louis, MO 63110, USA; alurij@wustl.edu (J.A.); cooper_m@wustl.edu (M.A.C.)

**Keywords:** TLR, immune system, inborn errors of immunity

## Abstract

Toll-like receptors (TLRs) are pattern recognition receptors that play a central role in the development and function of the immune system. TLR signaling promotes the earliest emergence of hematopoietic cells during development, and thereafter influences the fate and function of both primitive and effector immune cell types. Aberrant TLR signaling is associated with hematopoietic and immune system dysfunction, and both loss- and gain-of- function variants in TLR signaling-associated genes have been linked to specific infection susceptibilities and immune defects. Herein, we will review the role of TLR signaling in immune system development and the growing number of heritable defects in TLR signaling that lead to inborn errors of immunity.

## 1. Introduction

Toll-like receptors (TLRs) are a family of pattern recognition receptors that play a significant role in the development and maintenance of the immune system. These receptors recognize a wide variety of pathogens, as well as endogenous ligands associated with cellular damage. Signaling through TLRs leads to the production of proinflammatory cytokines and other inflammatory response mediators. TLRs and their signaling pathway effectors are therefore critical to the function of both the innate and adaptive immune system. In this review, we will discuss the role of TLRs in the development and maintenance of the immune system, focusing on the evolving body of literature linking heritable defects in TLR signaling to specific inborn errors of immunity (IEI).

## 2. Overview of TLR Signaling

Toll-like receptors (TLRs) are a family of pattern recognition receptors that play a central role in the immune response to infection and cellular damage. They are type I transmembrane proteins that contain a ligand-recognition leucin-rich repeat (LRR) domain, a transmembrane domain, and a Toll/interleukin 1 receptor (TIR) homology domain [1]. There are 10 TLR family members in humans, and 12 in mice [2,3,4,5,6,7,8]. Some of the TLRs, including TLR1, TLR2, TLR4, TLR5, TLR6 and TLR11, are localized to the plasma membrane, while others, including TLR3, TLR7, TLR8 and TLR9, are found in endosomes. Most function as homodimers, with the exception of TLR2, which heterodimerizes with TLR1 or TLR6 [9,10,11]. TLRs recognize foreign pathogen-associated molecular patterns (“PAMPs”), as well as endogenous by-products of cellular damage, or so-called damage-associated molecular patterns (“DAMPs”) [12], and are expressed in a wide variety of hematopoietic and non-hematopoietic cells, including effector immune cells (e.g., dendritic cells, macrophages, lymphocytes, granulocytes), hematopoietic stem and progenitor cells (HSPCs), and non-immune populations such as epithelial and endothelial cells [13,14,15,16,17,18,19,20]. TLR family members each have a unique cadre of natural ligands (PAMPs and DAMPs). In general, the TLRs expressed on the cell surface recognize microbial membrane lipids, proteins and lipoproteins, and the endosomal TLRs recognize both microbial- and self-derived nucleic acids [21].

Signaling through the TLRs involves the recruitment of intracellular adaptor proteins, which ultimately lead to the activation of transcription factors such as nuclear factor κ-light-chain enhancers of activated B cells (NF-κB), activating protein-1 (AP-1), interferon regulatory factor 3 (IRF3) and IRF7, and the production of proinflammatory cytokines (Figure 1). Most TLRs, with the exception of TLR3, utilize the intracellular signaling adaptor myeloid differentiation primary response gene 88 (MyD88) [22]. Activated TLRs recruit MyD88, followed by members of the serine-threonine kinase interleukin-1 receptor-associated kinase (IRAK) family (including IRAK1, IRAK2 and IRAK4) [23,24,25]. Together, these proteins form the “Myddosome.” The E3 ubiquitin ligase tumor necrosis factor (TNF) R-associated factor 6 (TRAF6) is then recruited to this complex, leading to its subsequent activation and stimulation of transformation growth factor beta-activated kinase 1 (TAK1), followed by activation of the NF-κB and mitogen-activated protein kinase (MAPK) pathways and the production of pro-inflammatory cytokines such as interleukin-1 (IL-1), IL-6, IL-8, tumor necrosis factor alpha (TNFα) and others [26]. Endosomal TLRs, including TLR3, TLR7, TLR8 and TLR9, act via the Myddosome and TRAF6 to stimulate NF-κB and IFR7. TLR3 utilizes the adaptor TIR-domain-containing adaptor-inducing interferon-β (TRIF) instead of MyD88 [27], and TLR4 signals via both MyD88- and TRIF-dependent pathways [28]. TRIF binds to TNF receptor-associated factor 3 (TRAF3), which then recruits the IKK-related kinases TANK-binding kinase 1 (TBK1) and IKKε, thus activating IRF3 and stimulating the production of type I interferons (IFNs) [29]. Additionally, TRIF interacts with TRAF6 and promotes the activation of NF-κB and MAPKs [30].

## 3. TLR Signaling in Immune System Development and Maintenance

TLRs regulate both the development and function of the immune system. Several studies have found that TLR signaling promotes the emergence of HSPCs from the hemogenic endothelium during embryogenesis in mouse and zebrafish [31,32]. Specifically, stimulation of TLR4 in hemogenic endothelial cells promotes HSPC development via Notch activation, and loss of TLR4 signaling leads to a significant reduction in HSPC emergence [31]. Thus, TLR signaling regulates immunity from the earliest steps of immune system development. Thereafter, TLR signaling helps to instruct the fate and function of both primitive and effector immune cells. In HSPCs, TLR signaling promotes proliferation and myeloid differentiation. Treatment of mice or stimulation of HSPCs in culture with TLR agonists, for example, leads to an expansion of HSPCs and differentiation along the myeloid lineage [20,33]. Notably, chronic exposure to TLR ligands, while expanding immunophenotypic HSPCs, leads to a loss of their function (repopulating and self-renewal activities) [34]. These effects of TLR signaling on HSPC proliferation and function are mediated by both cell-autonomous and non-cell autonomous mechanisms [35,36,37]. In mature effector immune cells, TLR signaling is central to the initiation and development of both the innate and adaptive responses to infection and injury. As noted above, TLR signaling leads to the production of type 1 interferons and other proinflammatory cytokines, and also promotes the production and secretion of other antimicrobial factors such as nitric oxide and defensins that promote pathogen killing [38]. TLR stimulation on antigen presenting cells upregulates the expression of costimulatory molecules (CD40, CD80, and CD86) and IL-12 to facilitate the differentiation of naïve T cells into antigen-specific Th1 effector T cells [39].

Aberrant TLR signaling is associated with susceptibility to infectious diseases, as well as to autoimmune diseases [40,41], chronic inflammatory diseases [41,42], and cancer [43,44]. Herein, we will focus on the growing body of literature linking germline variants in TLRs and TLR signaling pathway effector genes to specific immunodeficiencies.

## 4. Inborn Errors of Immunity (IEI) Associated with Inherited Defects in TLR Signaling

A number of IEI have been associated with inherited defects in TLR signaling (Figure 1). In general, the degree of immunologic impairment (and therefore the types and severity of infections) differ based on where in the TLR signaling pathway the defects occur. More proximal loss-of-function defects, such as those found in TLR3 or in the adaptors MyD88 and IRAK-4, are associated with a restricted cadre of infections. Conversely, defects in more distal effectors of TLR signaling, such as those affecting NF-κB, have more broad and deleterious effects on the immune response and are associated with a wider array of infections and other phenotypes.

## 5. MyD88 and IRAK-4 Deficiency and IEI

As described above, most TLRs, with the exception of TLR3 (and to some extent, TLR4), signal via the adaptor MyD88, which then complexes with IRAK-4 to initiate the formation of the Myddsome. In addition to TLRs, the interleukin-1 receptors (IL-1Rs) utilizes MyD88 and IRAK4 for signaling [45,46]. Inherited defects in both MyD88 and IRAK4 have been described [47,48], with loss of either factor leading to a similar phenotype involving a predilection for severe bacterial infections in early childhood.

IEI resulting from inherited defects in MyD88 were initially described in nine children from five unrelated kindreds with recurrent, severe pyogenic bacterial infections [47]. Four of the patients were found to have a homozygous in-frame deletion in *MYD88* (E52del), one had compound heterozygous missense variants (L93P; R196C), and two had homozygous missense variants (R196C). Functional testing of these alleles in patient-derived cell lines found them to be unresponsive to IL-1β and IRAK-4/MyD88-dependent TLR agonists. Furthermore, peripheral blood mononuclear cells from patients with MyD88 deficiency fail to produce cytokines (e.g., IL-1, IL-6) when stimulated with such agonists, suggesting that these variants are loss-of-function (LOF). Responses to poly(I:C) (TLR3 ligand) were largely normal. Inherited deficiency of IRAK-4 phenocopies that of MyD88 deficiency, and was initially described in three unrelated children with recurrent streptococcal and staphylococcal infections [48]. One patient had a homozygous deletion in exon 7 of IRAK-4 (821delT), and the other two patients had a single base-pair nonsense variant resulting in a premature stop codon (Q293X). As with the MyD88 variants, functional studies confirmed these to be LOF alleles resulting in IRAK-4 deficiency.

Deficiencies in both MyD88 and IRAK-4 are characterized by recurrent, severe pyogenic bacterial infections involving *Streptococcus pneumoniae*, *Staphylococcus aureus* and *Pseudomonas aeruginosa* [47,48,49,50]. Infections first occur early in life, usually before the age of two years, and include cellulitis, meningitis, arthritis, skin abscesses and sepsis. Systemic signs of inflammation (e.g., plasma C-reactive protein levels, fever) are often weak or delayed. As discussed above, both syndromes involve loss of protein expression and/or function and are inherited in an autosomal recessive pattern. Notably, affected individuals are otherwise healthy, with clinically normal resistance to other microbes including viruses, fungi, parasites and other bacteria. Furthermore, while MyD88- and IRAK-4-deficient children are at risk of life-threatening invasive bacterial infections early in life, with a mortality rate of approximately 38%, the susceptibility to infection decreases significantly with age. No infectious deaths have been reported in these patients after the age of 8 years, and no invasive infections have been reported after the age of 14 years (including patients not on antibiotic prophylaxis) [47,48,49,50]. Other features of MyD88- and IRAK-4- deficient patients include normal serum immunoglobulins, low-normal to normal specific antibody levels to polysaccharide antigens, normal B cell subsets and function in vitro, and normal T cell responses to mitogens and recall antigens. They may also have peripheral eosinophilia and elevated serum IgE [50].

The clinical diagnosis of MyD88 or IRAK-4 deficiency is best made by genetic sequencing. Functional testing of TLR stimulation is challenging, and multiple factors including sample handling can alter clinical testing results. Patients are generally treated with antibiotic prophylaxis for bacterial infection. In addition, for those (about 50% of patients) without adequate responses to the polysaccharide pneumococcal vaccine, immune globulin replacement therapy may be helpful [49].

The relatively narrow susceptibility to infection in humans contrasts with MyD88-deficient mice, which are much more broadly vulnerable to bacterial, viral, fungal and parasitic infections under experimental conditions (reviewed in reference [51]) [51]. While the reason for this discrepancy is not clear, it could be due to differences in experimental versus natural acquisition of infection, as well as differences in the biology of the immune system between mice and humans [52]. The fact that patients with MyD88 and IRAK-4 deficiency do not suffer from a wider variety of infections suggests that additional innate mechanisms exist to compensate for their loss in the detection of pathogens, and emphasizes the importance of identifying and studying such defects of the immune system in patients. Furthermore, the fact that the infections that they do acquire are largely confined to early childhood suggests that maturation of the innate and/or adaptive immune system diminishes the reliance on these factors for pathogen resistance with age.

## 6. TLR3 Pathway Variants and IEI

TLR3 is an endosomal TLR that binds to double-stranded RNA (dsRNA), which is produced during the replication of many viruses, and TLR3 signaling is thus important for the immune response to certain viral infections. TLR3 is expressed on myeloid dendritic cells and macrophages, as well as non-immune cells including neurons, astrocytes, microglia, fibroblasts and a variety of epithelial cells [53,54,55,56,57,58,59,60,61,62]. As detailed above, TLR3 signals via TRIF, which bind to TRAF3, recruit the IKK-related kinases TBK1 and IKKε, and ultimately activate IRF3 and IRF7 to stimulate the production of type 1 IFNs [27,29]. TRIF also interacts with TRAF6, promoting the MyD88-independent activation of NF-κB and MAPKs [30]. Given its role in viral recognition and the production of IFN, individuals with deficiencies in TLR3 signaling are susceptible to certain viral infections. Specifically, infections with HSV-1, influenza, and, most recently, SARS-Co-V2 have been attributed to pathogenic germline variants in TLR3 pathway genes.

## 7. TLR3 Pathway Variants and Herpes Simplex Encephalitis

Germline variants in TLR3 pathway genes including TLR3 [63,64,65,66,67], UNC93B1 [68], TRIF [66,69], TRAF3 [70], TBK1 [66,71], and IRF3 [72], as well as two genes important for type I IFN signaling (IFNAR1 and STAT1), have been linked to increased susceptibility to herpes simplex virus encephalitis (HSE). Herpes simplex virus type I (HSV-1) is a double-stranded enveloped DNA virus that typically causes asymptomatic infection or benign self-healing infections (e.g., stomatitis, gingivitis, labialis). dsRNA occurs as an intermediate in the replication cycle of HSV-1, which is then recognized by TLR3. TLR3 is expressed on CNS-resident cells permissive for HSV-1 infection, including microglia, neurons and oligodendrocytes [73,74,75], and can invade the CNS via the olfactory bulb or trigeminal nerves causing forebrain or brainstem HSE. Upon infection (primary or reactivation) with HSV-1, activated TLR3 signaling leads to production of IFNs which then initiate an adaptive immune response. Lack of IFN production by infected cells allows for increased viral replication and cell death. Genetic susceptibility to HSE was suggested by a series of case reports of familial HSE [76,77,78,79]. Furthermore, a French study of children with HSE identified a high frequency of consanguinity in affected children, supporting an autosomal recessive susceptibility to this infection [80]. Notably, people with HSE and TLR3 pathway deficiency are usually resistant to other types of infections, including HSV-1-related diseases outside of the CNS.

The connection between TLR3 deficiency and HSE was first described by Zhang and colleagues from the Casanova laboratory [63], who identified a dominant-negative heterozygous variant in TLR3 (P554S) in two unrelated otherwise healthy French children with HSE. Interestingly, three relatives of these affected children were also found to be heterozygous for the variant; however, they had not suffered from HSE in spite of HSV-1 seropositivity. Thus, the P554S TLR3 variant confers a predisposition to HSE with incomplete penetrance. This variant is located in a region thought to be important for ligand binding and TLR dimerization. Fibroblasts from individuals bearing the P554S variant displayed impaired NF-κB and IRF3 activation and cytokine responsiveness to the TLR3 ligand polyinosine-polycytidylic acid (poly(I:C)), as well as defective IFN-dependent control of viral replication and enhanced cell mortality upon infection with HSV-1. While monocyte-derived dendritic cells, NK cells and CD8T cells from affected individuals similarly displayed impaired response to poly(I:C), their blood DCs and keratinocytes responded normally. Furthermore, the peripheral blood mononuclear cells of affected individuals produced cytokines normally in response to most other viruses. Thus, TLR3-independent pathways appear to prevent the dissemination of HSV-1 outside of the CNS and provide resistance to most other viruses in TLR3-deficient patients [80]. Since this initial discovery of a TLR3 germline variant is associated with HSE, several additional variants conferring a loss of TLR3 activity and susceptibility to HSE have been described [64,65,66,67].

TLR3 is an endosomal TLR, and in resting cells co-localizes with UNC93B, a transmembrane protein that is important for transportation of all endosomal TLRs [81]. Variants in UNC93B therefore lead to defective signaling via TLR3, TLR7, TLR8 and TLR9. While UNC93B-deficient mice are susceptible to murine cytomegalovirus as well as bacterial infections [82], humans with recessive deficiencies in UNC93B are typically affected only by HSV-1 (despite impaired TLR3, TLR7, TLR8 and TLR9 signaling) [68]. Casrouge et al. [68] described two unrelated patients, each born to first-cousin parents, who presented with HSE but no other unusual infection history. Peripheral blood mononuclear cells (PBMCs) from these patients stimulated with HSV-1 produced significantly lower levels of IFN-α and IFN-β, and marginally lower IFN-λ than cells from healthy control individuals. They also had impaired production of these cytokines in response to a host of other viruses. Patient cells displayed normal cytokine production to LPS, but impaired IFN-α, -β, -λ, IL-1β, TNF-α and IL-6 production in response to agonists of TLR7, TLR8 and TLR9. Patient 1 was homozygous for a 4 nucleotide deletion in exon 8 of UNC93B1 (1034del4), and patient 2 was homozygous for single nucleotide substitution at the last nucleotide of exon 6 (781 G > A). UNC93B-deficient fibroblasts infected with VSV or HSV-1 showed high rates of viral replication and cytolysis. Cell viability was rescued if cells were treated with recombinant INF-α–2b before viral infection, demonstrating that cell death results from enhanced viral growth in the face of impaired IFN-α/β production. These findings implicated UNC93B deficiency as the cause of HSE in these patients.

TLR3 signals via the adaptor TRIF, and both autosomal dominant (AD) and autosomal recessive (AR) deficiency of TRIF have been attributed to inherited predisposition to HSE. Sancho-Shimizu and colleagues [69] described two different unrelated children with variants in TRIF, including a homozygous nonsense variant (R141X) that results in a lack of detectable protein, and a heterozygous missense variant (S186L), that leads to the production of dysfunctional, hypomorphic protein. Notably, the S186L variant was also found in unaffected relatives, suggesting that this variant confers a predisposition to HSE with incomplete clinical penetrance. Fibroblasts from the patients with R141X and S186L variants failed to produce IFN-β, IFN-λ and IL-6 in response to poly(I:C), HSV-1 or VZV infection, and were susceptible to HSV-1 viral growth and cell death. Mork and colleagues [66] identified two additional unrelated patients, both adults, with HSE and heterozygous TRIF variants (A568T and S160F). The PBMCs from both patients displayed impaired IFN-β responses to HSV-1 infection, indicating that these are loss-of-function variants.

An autosomal dominant, de novo germline TRAF3 LOF variant was described in a young adult with a history of HSE in childhood [70]. The variant (R118W) was associated with reduced TRAF3 protein expression and impaired fibroblast cytokine production (IFN-β, IFN-λ and IL-6) in response to poly(I:C). The R118W allele was found to exert a dominant-negative effect on overall TRAF3 protein expression, possibly via inhibition of trimer formation and the promotion of protein instability. In addition to TLR3-TRIF, TRAF3 interacts with various TNF receptors, influencing signaling via pathways such as CD40, LT-βR, and BAFFR [83,84]. Indeed, R118W mutant monocyte-derived dendritic cells and B cells showed impaired production of cytokines in response to activation with CD40L, and patient-derived fibroblasts were resistant to lymphotoxin-B receptor (LT-βR)-induced cell death. Of note, the R118W variant has also been described as a somatic variant in multiple myeloma [85,86]. In spite of the experimental effect of this variant on multiple pathways, the patient was clinically healthy, off antimicrobial prophylaxis, with the exception of a history of HSE.

The binding of TRIF and TRAF3 recruits the TBK1 and IKKε kinases, which subsequently activate IRF3 and NF-κB. Two heterozygous variants in TBK1 (D50A and G159A) were described in two unrelated children with HSE [71]. Both alleles were associated with autosomal-dominant inheritance, but via different mechanisms: the haploinsufficient D50A allele confers protein instability, and the dominant negative G159A allele displays impaired kinase activity. Both alleles sensitized fibroblasts to viral (HSV-1 and VZV) infection, and were rescued by IFN-α–2b treatment. A third variant (I207V) was identified in a 50-year-old previously healthy female with HSE. PMBCs from this patient displayed impaired CXCL10 and TNF-α production in response to HSV-1 infection, but increased induction of IFN-β in response to poly (I:C) [66].

A heterozygous LOF variant leading to haploinsufficiency of IRF3 was described by Andersen et al. [72] in an adolescent female with HSE. The variant (R285Q) interfered with IRF3 phosphorylation and dimerization, resulting in deficient IRF3 transcriptional activity and impaired IFN response of fibroblasts and PBMCs to synthetic TLR3 agonists and HSV-1 infection. The patient’s father was noted to be a healthy carrier of the same variant, indicating incomplete penetrance. A second heterozygous IRF3 variant (A277T) that similarly impaired PBMC response to poly (I:C) and HSV-1 infection was detected in a 34- year-old with HSE [66].

## 8. TLR3 Pathway Variants and Influenza

In the past several years, inherited defects in TLR3 and type 1 IFN signaling have been linked to severe influenza A virus (IAV) infection in work from the Casanova laboratory [87,88,89,90]. Lim et al. [88] identified three unrelated children with influenza A-associated acute respiratory distress syndrome (ARDS) and heterozygous TLR3 variants leading to AD TLR3 deficiency. Two of these patients carried the P554S variant, which was previously associated with germline susceptibility to HSE [63]. The third variant (P680L) is a loss-of-function variant that results in misfolded, uncleaved protein that is largely retained in the ER [88], and which also is deficient in dimerization and ligand binding [91,92]. Five healthy relatives of these affected children were found to carry the variants as well, indicating incomplete clinical penetrance for severe influenza A infections. Notably, while TLR3 mutant fibroblasts and the patients’ iPSC-derived pulmonary epithelial cells (PECs) displayed enhanced susceptibility to IAV infection and reduced IFN-β and IFN-λ production, their peripheral blood leukocytes produced normal amounts of IFNs in response to IAV. As pulmonary epithelial cells are a primary target of IAV, specific loss of IFN-β and IFN-λ upon IAV infection likely underlies the pathogenesis of severe IAV-ARDS in TLR3-deficient patients. Indeed, pre-treatment of TLR3 mutant PECs with IFN-α-2b or IFN-λ rescued their vulnerability to IAV infection [88]. Ciancanelli and colleagues [89] reported on a 7-year-old girl with life-threatening ARDS from pandemic H1N1 2009 influenza A virus who was found to have compound heterozygous IRF7 variants leading to a loss of IRF7 function and impaired production of type I and III IFNs in response to influenza infection. Interestingly, there are no reports to date of patients suffering from both IAV and HSE, despite the overlap in genetic etiologies [87].

## 9. TLR3 Pathway Variants and COVID-19

Zhang and colleagues [93] considered that similar variants in TLR3 and IFN production or response may underlie susceptibility to lethal coronavirus disease 2019 (COVID-19). Indeed, they reported that inborn errors of TLR3- and IRF7-dependent type IFN production predispose individuals to life-threatening COVID-19 pneumonia. Specifically, they identified LOF variants governing TLR3- and IRF7-dependent type I IFN production in 3.5% of a group of 659 hospitalized COVID-19 patients, with a significant enrichment in these variants in individuals with life-threatening pneumonia compared to a control group with asymptomatic or benign infection. Eight different genes were represented by these variants, including TLR3, TRIF, UNC93B1, TBK1, IRF3, IRF7, IFNAR1 and IFNAR2. Among these, both autosomal recessive and autosomal dominant inheritance patterns were found, including known (AR IRF7 and IFNAR1 deficiencies, and AD TLR3, TRIF, TBK1 and IRF3 deficiencies) and novel (AD UNC93B, IRF7, IFNAR1 and IFNAR2 deficiencies) disorders of these pathways. As predicted, plasma levels of IFN-α were significantly lower in patients with these genotypes compared to those without during an acute phase of COVID-19. Finally, the authors demonstrated experimentally that plasmacytoid dendritic cells from IRF7-deficient patients failed to produce type 1 IFN in response to SARS-CoV-2 infection, and patient-derived fibroblasts deficient for TLR3, IRF7 or IFNAR1 are susceptible to SARS-CoV-2 infection in vitro. Notably, two SARS-CoV-2-infected patients with known inborn errors of TLR3 and IRF3 were successfully treated with IFN-α–2a [94]. In both cases, the patients presented at the early stage of clinical manifestations with high viral loads based on nasal swab PCR, and were given a single dose of Peg-IFN-α–2a. Both patients had a rapid resolution of symptoms following IFN administration, suggesting that early treatment with exogenous type 1 IFN may benefit SARS-CoV-2-infected patients with known defects in TLR3- and IRF7-dependent IFN production.

TLR3 pathway defects thus predispose humans to a narrow collection of viral infections. Notably, despite the overlap in the types of variants associated with different viral susceptibilities, most patients do not present with more than one type of infection. Furthermore, incomplete penetrance was observed in multiple cases, with clinically unaffected relatives carrying pathogenic variants. Thus, while TLR3 signaling variants confer an increased risk of certain viral infections, redundancy within the immune system for viral detection and other mitigating factors (e.g., age of exposure, variants within other immune related genes) likely restrict the clinical phenotype.

## 10. TLR8 Gain-of-Function Variants and IEI

Toll-like receptor 8 is an endosomal pathogen sensor that recognizes single-stranded RNA including viral ssRNA such as Influenza [95], Sendai [96], and Coxsackie B [97] virus and bacterial RNA such as *Mycobacterium bovis* [98] and *Helicobacter pylori* [99]. It can also detect synthetic oligoribonucleotides or chemical analogs such as imidazoquinolines [100]. The gene encoding TLR8 is located on the X-chromosome and there are two splice variants of TLR8—TLR8v1 and TLR8v2—of which TLR8v2 is the most conserved form of TLR8 expressed in human cells [98]. TLR8 is primarily expressed in monocytes, macrophages, myeloid dendritic cells and neutrophils and, like TLR3, requires UNC93B1 for its endosomal targeting [101] where it exists as a pre-formed dimer. It possess a leucine-rich repeat (LRR) region, a transmembrane domain, and a Toll–IL-1 receptor homology (TIR) domain [102]. Ligand binding induces a conformational change in the TIR dimer interface that causes the cytoplasmic domains to contact each other to initiate the TLR8-MyD88 signal transduction pathway. This leads to NF-kB activation, IRF-7 and IRF-5 activation via the IRAK pathway. This subsequently leads to production of pro-inflammatory cytokines and type I IFNs. TLR8 is one of the least-studied members of the TLR family, primarily due to a lack of small animal models. For instance, mouse TLR8 lacks a five amino-acid residue motif that is highly essential for human TLR8 ligand binding and function [103], hence murine TLR8 does not recognize ligands that activate human TLR8. Murine TLR8 is functional and mice deficient in TLR8 have increased TLR7 signaling due to the lack of inhibition by TLR8, and DCs produce high amounts of cytokines causing spontaneous autoimmunity, autoantibodies, splenomegaly, and reduced B cell numbers [104].

Recently, we described germline and somatic variants in TLR8 as an underlying monogenic cause of IEI in patients with recurrent infections, neutropenia, lymphoproliferation, hypo-gammaglobulinemia, and bone marrow failure [105]. In our cohort of six unrelated male patients, five patients harbored somatic mosaic variants in TLR8, with four patients having the same mosaic variant (P432L) and the fifth patient with a different mosaic variant (F494L). TLR8 mosaicism was detected at similar allelic frequencies in DNA samples from whole blood, fibroblast, saliva and/or sorted immune subsets in 4/5 patients, indicating that the post-zygotic mutational event leading to mosaicism occurred at an early stage of embryonic development in these patients. Patients with mosaicism had less than 30% cells harboring the variant in their peripheral blood samples (range, 8–26%), suggesting that the variants exert a “dominant” phenotype. The sixth patient harbored a de novo germline variant and died at a young age due to severe fungal infections associated with refractory neutropenia. Functional testing of the variants in a TLR8-deficient NF-κB reporter cell line revealed that all the variants result in a gain-of-function (GOF) phenotype.

We hypothesize that TLR8 GOF variants lead to a hyperinflammatory state and immune dysregulation. Supporting this, analysis of serum cytokines demonstrated significantly increased TNFα, IL-1β, IFNγ, BAFF, IL-2Rα, IL-12/23 p40 and IL-18. Immunological evaluation revealed an inverse CD4:CD8 ratio, a skewed CD45RA/CD45RO ratio, a high percentage of CD8+ TEMRA cells in multiple patients, presence of a dominant T cell clone in one patient and a T-cell large granular lymphocyte (T-LGL) phenotype in one patient, suggesting an activated T cell phenotype. Multiple patients had low B cell numbers, with reduced class-switched memory B cells, suggesting a B cell maturation defect. The functional consequence of the GOF variants on patient-primary cells with mosaicism was established using patient-specific iPSCs. Differentiation of myeloid cells from patient-specific iPSCs identified cells with the variant as having increased phosphorylation of NF-κB to low doses of TLR8 stimulation. Additionally, enhanced production of pro-inflammatory cytokines like IL-6, TNF-α and IL-1β was also observed, supporting the presence of cytokine-driven mechanisms of disease pathogenesis.

Patients with TLR8 GOF were all relatively refractory to standard therapy for neutropenia, including G-CSF treatment. All patients were trialed on various immune suppression agents, and three patients required hematopoietic stem cell transplantation (HSCT) due to evolving bone marrow failure. The optimal therapy for these patients is still unclear, but may include inhibiting cytokine signaling and ultimately HSCT.

Notably, this recent discovery of mosaic, GOF TLR8 variants associated with severe immune deficiency and multiple cytopenias suggests that while the immune system can largely compensate for the loss of individual TLRs (including multiple TLRs simultaneously, as with MyD88 or IRAK-4 LOF), sustained hyperactivation of TLR signaling can have more broadly deleterious effects on immune system function. In addition, mosaic variants may be missed on standard exome and genome analyses, begging the question of whether additional such variants will be identified in the future to account for previously unexplained cases of IEI. Moving forward, sequencing of patients with immunodeficiency of an unknown etiology should consider mosaic variants (e.g., those with low allele frequencies) as potentially pathogenic.

## 11. IKK Complex and NFKBIA (IKBA) Variants and IEI

NF–κB is important for multiple innate and adaptive immune pathways (e.g., TLRs, nucleotide-binding oligomerization domain (NOD)-like receptors, T- and B-cell receptors, CD40 and others), as well as developmental pathways (e.g., receptor activator of NF-κB in osteoclasts and ectodysplasin A in ectoderm). NF-κB activity is regulated by NF-κB inhibitor proteins alpha and beta (NFKBIA and NFKBIB), also known as inhibitor of kappa B alpha and beta (IKBα and IKBβ). These inhibitors sequester NF-κB in the cytoplasm, and upon their degradation after phosphorylation by IKK (a heterotrimer of alpha, beta and gamma chains), NF-κB can enter the nucleus. Variants that effectively reduce NF-κB activity, including hypomorphic variants in the IKK beta chain (encoded by IKBKB) or the IKK gamma chain (also known as NF-κB essential modifier, or NEMO, encoded by the IKBKG gene on the X chromosome), or activating variants in NFKBIA, cause a combined immunodeficiency with associated bone and skin findings.

Complete LOF variants of NEMO result in embryonic lethality in males and a severe genodermatosis known as incontinentia pigmenti in females. Hypomorphic variants, however, result in X-linked anhidrotic ectodermal dysplasia with immunodeficiency (XL-EDA-ID). Approximately 100 patients with 43 different NEMO variants have been reported, all of which impair NF-κB signaling [106,107,108]. These variants have a varying impact on NEMO function, and are therefore associated with diverse clinical and immunologic phenotypes [109]. All patients have some degree of immune dysfunction, with variable susceptibilities to pyogenic bacterial, mycobacterial, fungal and viral infections. Most (though not all) patients with NEMO deficiency display features of EDA (including skin abnormalities, hypoplastic or absent sweat glands, sparse hair, tooth abnormalities and nasal or forehead dysmorphology), and they may also present with autoimmunity and inflammatory conditions, particularly inflammatory bowel disease [109].

An autosomal dominant form of EDA-ID (AD-EDA-ID) results from GOF variants in the gene that encodes the IkBα subunit (NFKBIA), preventing its phosphorylation and subsequent degradation and thus enhancing its activity to inhibit the nuclear translocation of NF-κB [110,111]. Eleven different variants have been identified in 14 unrelated patients [110,112,113,114,115,116,117,118,119,120,121], including missense variants affecting phosphorylation sites (S32, S36, or neighboring residues), and nonsense variants upstream from S32 associated with re-initiation of translation downstream from S36. Patients suffer from severe pyogenic, mycobacterial, fungal and viral infections beginning in infancy. They all display severe B-cell deficiency with impaired antibody production, as well as varying degrees of other features including lymphocytosis, dysfunctional α/β T cells, lack of circulating γ/δ T cells, and lack of peripheral lymph nodes. Finally, almost all patients with these GOF NFKBIA variants display features of EDA, including absent or dysfunctional sweat glands, abnormal teeth, and sparse hair. Of the fourteen known patients described by Boisson and colleagues in 2017 [111], 11 had received HSCT. Six of these transplanted patients died (three from bacterial sepsis, one from neurodegenerative disease, one from acute respiratory distress and one from cerebellar hemorrhage), and 5 were successfully transplanted, 3 of which have persistent partial immunodeficiency [111].

Cuvelier et al. [122,123] described a series of 16 patients with homozygous IKBKB variants (1292dupG), resulting in a complete loss of IKKβ expression, that presented in early infancy with severe bacterial, viral, fungal and mycobacterial infections. Unlike individuals with IKK/NEMO variants, none of these patients exhibited ectodermal dysplasia. Thymic hypoplasia, minimal lymph node and tonsillar tissue, and an absence of splenic germinal follicles were noted in all patients. T and B cells, while quantitatively normal, were phenotypically naïve, and the patients had hypogammaglobulinemia. Eight of the affected infants underwent HSCT, and the other eight died of overwhelming infection before HSCT could be performed. Of those transplanted, 3 remained alive 6 months, 6 years and 7 years post-transplant. Two patients died from complications of infections present at the time of the transplant and three died of new infections acquired post-transplant (two in association with graft failure).

## 12. TLR Pathway Polymorphisms and Disease

In addition to the monogenic disease discussed in the sections above, numerous single-nucleotide polymporphisms (SNPs) in TLR pathway genes with more subtle effects on protein function have been associated with altered susceptibility to infection and inflammatory diseases. Discrepancies exist in the literature regarding the impact of various TLR-related SNPs on disease, and their influence on the immune system often depends on environmental factors, sex, ethnic background and the presence of polymorphisms in other immune-related genes. A comprehensive review of clinically significant TLR polymorphisms is beyond the scope of this review, and may be found in several other excellent publications [124,125,126]. Herein, we will briefly review just a few of the more commonly studied TLR polymorphisms associated with infection susceptibility.

A number of studies have identified polymorphisms in TLRs that are associated with increased risk of severe fungal infections [127,128,129,130,131]. In a study of 338 patients with severe invasive candida infection (candidemia) and 351 non-infected controls, Plantiga and colleagues [127] found 3 TLR1 SNPs (R80T, S248N and I602S) that were significantly associated with susceptibility to candidemia, particularly in white individuals. PBMCs from patients carrying these SNPs displayed reduced cytokine production upon incubation with the specific TLR1/2 agonist PAM_3_CSK_4_. In a separate study [132], the TLR1 602S variant was shown to impair trafficking on TLR1 to the cell surface, and monocytes homozygous for this variant produced significantly less TNF-α in response to PAM_3_CSK_4_ compared to those with at least one 602I allele. Interestingly, the 602S allele, which is common in Caucasians, was found to be underrepresented in a population of 57 Turkish leprosy patients compared to 90 asymptomatic controls recruited from areas where this disease is endemic. Specifically, the homozygous 602S genotype was associated with a significant reduction in leprosy incidence (odds ratio of 0.48), suggesting that reduced TLR1 surface expression and function is protective against symptomatic leprosy.

Polymorphisms in TLR2 have been associated with a risk of various infectious diseases. The R677W and R753Q SNPs, for example, have been linked to impaired TLR2 signaling and enhanced susceptibility to tuberculosis infection. Consistent with its known role in viral recognition, TLR3 polymorphisms have been linked to susceptibility to viral infection. The L412F SNP, for example, has been associated with resistance to HIV infection in an Italian population [133]. Polymorphisms in TLR4 have been variably linked to increased risk of sepsis infection with gram negative bacteria, RSV bronchiolitis and disseminate candidiasis. A common stop codon SNP in the ligand-binding domain of TLR5 (329STOP) that is unable to mediate flagellin signaling has been associated with susceptibility to Legionella pneumophila pneumonia [134]. Finally, polymorphisms in TLR7 and TLR8 have been linked to susceptibility to HCV infection [135,136].

## 13. Discussion

TLRs are key pathogen-sensing receptors that play a central role in both the development and activation of the immune system. Not surprisingly, dysfunctional TLR signaling predisposes to infections, and a growing number of single-gene variants in the TLR pathways have been associated with inherited immune system defects. Most of the known TLR-associated immune deficiencies involve constitutional LOF variants in TLR pathway genes, and, with the exception of the IKK complex and IKBα variants, confer a relatively narrow susceptibility to infection with minimal effects on the immune system overall. Furthermore, many of the identified variants associated with infection susceptibility are not fully clinically penetrant. The reason for this narrow susceptibility and limited penetrance is not clear. Further research is needed to better understand the role of TLR signaling in different cells of the immune system, to identify redundancy within the TLR signaling pathways and among other pattern recognition receptors, and to understand how the roles of TLRs change with maturation and aging of the immune system. Furthermore, the cell-autonomous and cell non-autonomous effects of enhanced TLR signaling (e.g., in the case of TLR8 GOF variants) on the development and function of the immune system awaits further study. Finally, while studies in mice over the years have provided an invaluable insight into the role of TLR signaling in the immune system, the discrepancies in phenotypes between TLR signaling-deficient mice and humans (e.g., the much more broad infection susceptibility in MyD88-deficient mice compared to humans with MyD88 LOF variants) highlights the importance of utilizing patient samples whenever possible to help answer the outstanding questions regarding the role of TLR signaling in pathogen recognition and immune system function.

## Figures and Tables

**Figure 1 cells-10-01374-f001:**
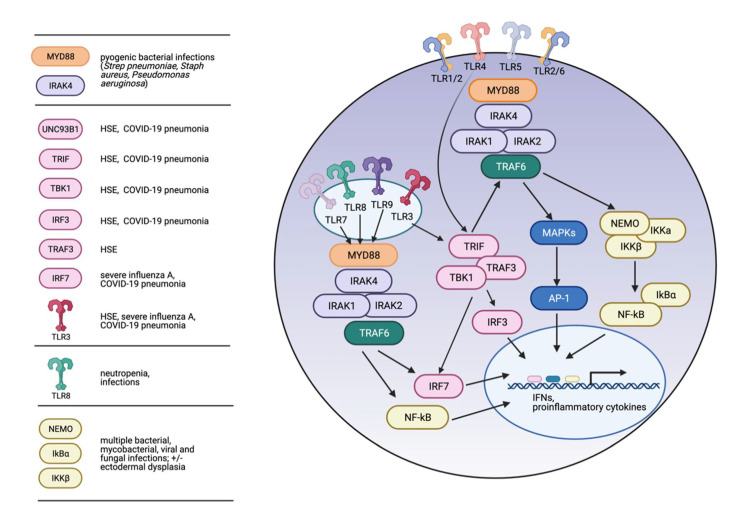
TLR pathway variants and immune deficiencies. TLRs signal via MyD88-dependent or TRIF-dependent pathways. Activated MyD88 recruits IRAKs1/2/4, then TRAF6, and ultimately activates MAPKs and NF-κB resulting in the production of proinflammatory cytokines. Activated TRIF (via ligation of TLR3 or TLR4) binds to TRAF3 and TBK1, ultimately activating IRF3 and IRF7 to stimulate the production of interferons. TRIF also interacts with TRAF6 to stimulate MyD88-independent activation of MAPKs and NF-κB. Highlighted on the left are TLR signaling pathway players in which variants have been associated with specific inborn errors of immunity, as indicated. Created with BioRender.com accessed on 27 April 2021.

## Data Availability

Not applicable.

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
