# Peer review of "Toll-Like Receptor Signaling in the Establishment and Function of the Immune System"

_cells, 2021, doi:10.3390/cells10061374_

Round 1

Reviewer 1 Report

1. Before resubmission, authors should ask native English speaker to edit their manuscript and check spelling errors. 

Author Response

The manuscript was written and edited by highly experienced and published investigators very competent in the English language. If the reviewer has specific concerns that they would like to point out in the text we would be happy to address them. 

Reviewer 2 Report

This review describes linkages between aberrant TLR signaling and hematopoietic or immune system dysfunction.  The authors describe numerous known loss- and gain-of-function variants.  Overall, this is a well written review and will serve as an excellent resource for those in the field. 

Critiques

  1. In my pdf copy, Greek letters like alpha, beta, kappa, etc. were not present, although they appear to be intended.  The authors and publishers will need to take care to make sure these characters are included.
  2. This review would be greatly enhanced if the authors, at a minimum, included a summary table of variants that included each TLR, adaptor, and/or TLR pathway component described, the variants observed, and references where they can be found. Including additional information as the authors deem appropriate (e.g. brief description of effect) could be appreciated by readers as well.  If necessary, this table can be split into multiple tables, as the authors see fit. 
  3. Line 343: For TLR8, there is reference to an extracellular LRR although the authors note that this receptor is expressed on endosomes.  This may require clarification.       
  4. On lines 104, 162, 334, and 399, dashes and sometimes periods precede the heading. Unless they serve a purpose, they should be removed. 

Author Response

Reviewer 2: This review describes linkages between aberrant TLR signaling and hematopoietic or immune system dysfunction.  The authors describe numerous known loss- and gain-of-function variants.  Overall, this is a well written review and will serve as an excellent resource for those in the field. We thank the reviewer for their comments.

Critiques

  1. In my pdf copy, Greek letters like alpha, beta, kappa, etc. were not present, although they appear to be intended.  The authors and publishers will need to take care to make sure these characters are included. This has been corrected.
  2. This review would be greatly enhanced if the authors, at a minimum, included a summary table of variants that included each TLR, adaptor, and/or TLR pathway component described, the variants observed, and references where they can be found. Including additional information as the authors deem appropriate (e.g. brief description of effect) could be appreciated by readers as well.  If necessary, this table can be split into multiple tables, as the authors see fit. 

We appreciate this suggestion and agree that a table could be useful. However, we have avoided doing this given the large and rapidly expanding number of disease-associated variants. We tried to cover the most commonly-reported variants in the text, and included a figure covering the broad categories of known defects, but felt that generating an accurate table with all variants would be untenable.

3. Line 343: For TLR8, there is reference to an extracellular LRR although the authors note that this receptor is expressed on endosomes.  This may require clarification. We have removed the words extracellular and intracellular to avoid confusion.       

4. On lines 104, 162, 334, and 399, dashes and sometimes periods precede the heading. Unless they serve a purpose, they should be removed. These have been removed.

Reviewer 3 Report

The review by Aluri et al from the Schuettpelz lab on TLR signaling in establishment and function of immune system is a comprehensive take on mutations in TLR genes in humans and their impact on disease susceptibility to various pathogens of bacterial, viral and fungal origins. I only have minor suggestions.

  1. The authors should consider changing/modifying their title as they focus mostly on TLR mutations in humans and their impact on susceptibility to pathogens with the exception of TLR8. There is only one paragraph devoted to TLRs in establishment, development  and maintenance of immune system (line 69).
  2. Line 23: Too many conjunctions used in one sentence "and". Please break the sentence to make it easy to understand.
  3. Line 32: Not all TLRs are expressed on the surface. Therefore the word "extracellular" is not inclusive to endosomal TLRs.  Please modify.
  4. Line 121: Please use Streptococcal or Staphylococcal instead of short forms.
  5. Line 215: Should be "in" instead of "is"
  6. All the special characters designating alpha, beta etc are missing due to formatting issues. Please make sure appropriate characters are used in the final version.
  7. Line 283: The full-form of AD is missing. Please make sure all abbreviations are properly expanded  when they are mentioned the first time. 

Author Response

Reviewer 3: The review by Aluri et al from the Schuettpelz lab on TLR signaling in establishment and function of immune system is a comprehensive take on mutations in TLR genes in humans and their impact on disease susceptibility to various pathogens of bacterial, viral and fungal origins. I only have minor suggestions.

We thank the reviewer for their suggestions, which we have addressed as follows:

  1. The authors should consider changing/modifying their title as they focus mostly on TLR mutations in humans and their impact on susceptibility to pathogens with the exception of TLR8. There is only one paragraph devoted to TLRs in establishment, development  and maintenance of immune system (line 69). Although we do like the current title, we could change to “Toll like receptor signaling defects and inborn errors of immunity,” if the editor and reviewers would prefer.
  2. Line 23: Too many conjunctions used in one sentence "and". Please break the sentence to make it easy to understand. This has been done.
  3. Line 32: Not all TLRs are expressed on the surface. Therefore the word "extracellular" is not inclusive to endosomal TLRs.  Please modify. This has been done.
  4. Line 121: Please use Streptococcal or Staphylococcal instead of short forms. This has been changed.
  5. Line 215: Should be "in" instead of "is". This has been changed.
  6. All the special characters designating alpha, beta etc are missing due to formatting issues. Please make sure appropriate characters are used in the final version. This has been fixed.
  7. Line 283: The full-form of AD is missing. Please make sure all abbreviations are properly expanded  when they are mentioned the first time. This has been done (in line 244).